# Generation of Replacement Options in Text Sanitization

**Annika Willoch Olstad**
Language Technology Group
University of Oslo
Oslo, Norway
annikaol@ifi.uio.no

**Anthi Papadopoulou**
Language Technology Group
University of Oslo
Oslo, Norway
anthip@ifi.uio.no

**Pierre Lison**
Norwegian Computing Center
Oslo, Norway
plison@nr.no

## Abstract

The purpose of text sanitization is to edit text documents to mask text spans that may directly or indirectly reveal personal information. An important problem in text sanitization is to find less specific, yet still informative replacements for each text span to mask. We present an approach to generate possible replacements using a combination of heuristic rules and an ontology derived from Wikidata. Those replacement options are hierarchically structured and cover various types of personal identifiers. Using this approach, we extend a recently released text sanitization dataset with manually selected replacements. The outcome of this data collection shows that the approach is able to suggest appropriate replacement options for most text spans.

## 1 Introduction

Most texts contain Personally Identifiable Information (PII), which is information that can be used to directly or indirectly identify an individual. This raises privacy problems, as privacy frameworks such as GDPR (GDPR, 2016) enshrine the right of each individual to control the availability and sharing of their personal information.

Although full, GDPR-compliant anonymization is difficult to achieve (Weitzenboeck et al., 2022), it is often desirable to apply *text sanitization* techniques to mask (i.e. remove or replace) PII from a given text and thereby conceal the identity of the persons referred to in the document. Those PII can either correspond to *direct identifiers* (e.g. names, addresses, telephone numbers or social security identifiers) or take the form of so-called *quasi-identifiers* which are information that do not identify a person when seen in isolation, but may do so when combined together (Elliot et al., 2016). Examples of quasi-identifiers are postal codes, gender, age, employer or profession.

Most text sanitization approaches operate by (1) detecting text spans that convey PII and (2) replacing them with a default string such as '***' or a black box (Lison et al., 2021; Pilán et al., 2022). However, this considerably reduces the utility of the sanitized document. An alternative is to replace the detected text spans with more general values that are less risky from a privacy perspective, but remain more informative than a default string. For instance, *Drammen* may be replaced by *[city in Norway]*, *Telenor* by *[telecommunications company]* and *February 5, 2023* by *[2023]*.

The paper makes two contributions:

- An approach (illustrated in Figure 1) that generates suitable generalization options for different types of PII, based on heuristic rules and an ontology derived from Wikidata.

- *WikiReplace*, an extension of the dataset from Papadopoulou et al. (2022a) in which human annotators select for each text span the most suitable replacement among the possible alternatives produced by the above approach. The dataset is made freely available[1].

The paper focuses on the specific problem of generating replacement choices for text spans expressing PII. The problem of how those text spans should be detected and classified lies therefore outside the scope of this paper. This span detection can be implemented using various types of sequence labelling models, as detailed in Dernoncourt et al. (2017); Lison et al. (2021); Pilán et al. (2022)

The rest of the paper is constructed as follows. Section 2 describes previous work in this task, while Section 3 presents the replacement approach. In Section 4 we present the dataset and evaluate its quality. We conclude in Section 5.

---

[1] https://github.com/anthipapa/bootstrapping-anonymization

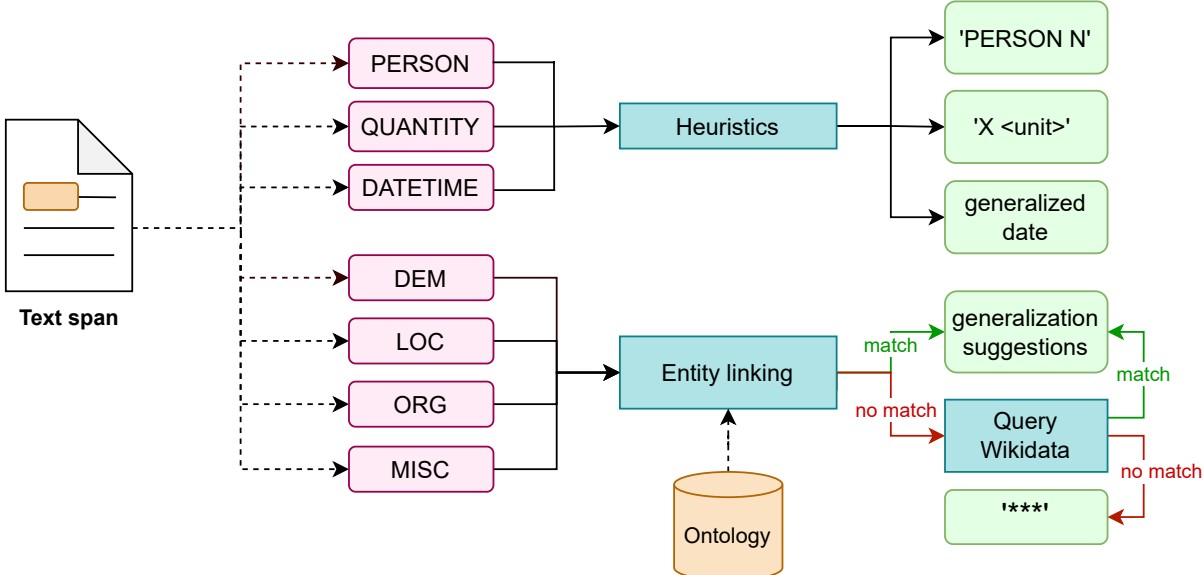

Figure 1: Generation of replacement options for text spans. Depending on the entity type, the replacements are produced using either heuristics or the Wikidata-derived ontology.

## 2 Related work

How to replace text spans expressing PII has been investigated in both Natural Language Processing (NLP) and in Privacy-Preserving Data Publishing (PPDP). Most NLP approaches (Bråthen et al., 2021; Dernoncourt et al., 2017; Pilán et al., 2022; Papadopoulou et al., 2022b) simply replace the detected text spans by a default string or a black box. Some alternatives include replacing text spans by pseudonyms (Dalianis, 2019; Volodina et al., 2020) or synthetic surrogates (Carrell et al., 2012). In the medical domain, identified names in patient records can be replaced with random names from a list (Dalianis, 2019). One can also rely on lexical substitution, in which target words are replaced with similar lexical entities, e.g. a synonym or hypernym (McCarthy and Navigli, 2007). This substitution can be implemented using various neural language models (Zhou et al., 2019; Arefyev et al., 2020).

Within the field of PPDP, the C-sanitize approach (Sánchez and Batet, 2016) frames the replacement of quasi-identifiers through an automatic sanitization process that mimics manual sanitization. It replaces identifiers with suitable generalizations, selected from a knowledge base, and an $a$ parameter that can be adjusted to trade between privacy protection and data utility. $t$-plausibility (Anandan et al., 2012) generalizes identifiers so that at least $t$ documents are derived through specialization of the generalized terms.

## 3 Generation of potential replacements

We follow the categorization of text spans expressing PII detailed in Pilán et al. (2022):

PERSON  Names of people.

CODE  Numbers and identification codes.

LOC  Places and locations.

ORG  Names of organizations.

DEM  Demographic attributes of a person, such as job title, education, ethnicity or language.

DATETIME  Specific date, time or duration.

QUANTITY  Quantity, including percentages or monetary values.

MISC  Every other type of information not belonging to the categories above.

Entities of type PERSON, QUANTITY and DATE-TIME are replaced using the heuristics in Section 3.1. Entities of type LOC, ORG, DEM and MISC are replaced by generalizations found in the ontology through entity linking, as described in Section 3.2.1. If no generalization can be found in the ontology, the system queries Wikidata directly. If this query does not return any generalization, '***' is returned. As entities of type CODE cannot be generalized, they are replaced by '***'.

### 3.1 Rule-based generation

We developed a set of heuristic rules to generalize entities of type PERSON, QUANTITY and DATETIME:

- **PERSON** entities are replaced by the text span [*PERSON N*], where N is an integer:

  "Ada Lovelace" → [*PERSON 1*]

  Terms that are found to refer to the same individual (based on e.g. coreference resolution) are assigned to the same integer.

- **QUANTITY** entities are replaced by X followed by the unit of measurement if applicable:

  "23 €" → [*X €*].

- **DATETIME** entities are generalized to the year, decade, or [*DATE*] as default value:

  "March 12, 1994" → [*1994*] or
  [*date in the 1990s*]
  "the following day" → [*DATE*].

Heuristics were chosen for these types of entities since they are usually not part of knowledge graphs that can be used to create ontologies.

### 3.2 Ontology-based generation

Entities of type LOC, ORG, DEM and MISC are generalized using an ontology. The ontology was constructed using Wikidata[2], a knowledge graph where pieces of information are linked together by *properties*. We consider specific membership properties, namely *instance_of* (P31), *subclass_of* (P279), *part_of* (P361), and *is_metaclass_for* (P8225), which express a hierarchical relation from specific to more general, as seen in Table 1.

| ID | Label | Example |
|---|---|---|
| P31 | instance_of | Oslo → capital city |
| P279 | subclass_of | capital city → city |
| P8225 | is_metaclass_for | genre → creative work |
| P361 | part_of | door → house |

Table 1: Wikidata properties employed to construct the generalization ontology.

The ontology contains all terms related to humans and their generalizations extracted using the properties mentioned above, with the addition of terms for *countries* and *nationalities*.

#### 3.2.1 Entity linking

The text span to generalize must first be linked to an appropriate term in the ontology. We first search for exact matches, followed by a *contained_in* search

in the ontology. Finally, if no entity is found, approximate string matching is employed to tentatively match the PII.[3].

If the above entity linking fails (which means that this term is absent from the ontology), we query Wikidata directly to get a match. If a match is found then we return the results, otherwise '***' is suggested as an appropriate replacement. Masking the term with '***' is both proposed when no match is found and as a final option for all PII, to provide an alternative when the provided generalization options are inappropriate.

#### 3.2.2 Ontology traversal

Every term related to a human in the ontology was used to fetch generalization options using the membership properties in Table 1. In the case of several available property options, the first one is selected. The ordering of properties added to the ontology was: P31, P279, P8225 and finally P361. Below is a list showing the generalizations of the term 'drummer' based on the *P31* property, and of the term 'mother' based on the *P279* property.

**drummer** → [percussionist] → [instrumentalist] → [musician] → [artist] → [creator] → [person] → ***

**mother** → [parent] → [first-degree relative] → [kin] → [person] → ***

The generalizations range from the most specific (most informative, but also potentially most risky in terms of identity disclosure) to the less specific (least risky, but also least informative).

## 4 Dataset

The dataset used for the data collection consists of 553 Wikipedia articles already annotated for text sanitization by Papadopoulou et al. (2022a). Wikipedia articles are suitable for this task as they are both dense in PII and publicly available. For each article, human annotators labelled the text spans that needed to be masked to protect the identity of the mentioned individual, while also retaining as much of the utility of the resulting text as possible. Each text span is also assigned to one of the 8 categories enumerated in Section 3.

---

[2]See `https://www.wikidata.org`. The dump file was downloaded on Sept. 13, 2022.

[3]A term is considered a match if the character-level edit distance is below a given threshold, set in our implementation to 15% of the length of the entity string.

| Type | Level 1 | Level 2 | L. > 2 | *** |
|---|---|---|---|---|
| DATETIME | 1025 | 1032 | 360 | 764 |
| DEM | 265 | 202 | 242 | 318 |
| LOC | 356 | 419 | 263 | 524 |
| MISC | 272 | 622 | 481 | 964 |
| ORG | 652 | 773 | 430 | 1066 |
| PERSON | 2478 | 85 | 0 | 18 |
| QUANTITY | 381 | 5 | 0 | 5 |
| **Total** | **5429** | **3138** | **1776** | **3726** |

Table 2: Levels of generalization per semantic type.

## 4.1 Annotation

We expanded the above dataset with the generalization options proposed by the system, and then recruited 9 annotators to select the most suitable replacement among the possible alternatives. To this end, we developed a web based annotation tool through which the annotators received a link to a web page containing the documents they were assigned to annotate. The annotators were also provided with annotation guidelines (see Appendix A 5). The annotators were required to select exactly one option per marked text span. Each annotator annotated 81 documents, whereof 59 were randomly selected, with the remaining 22 documents being multi-annotated. Two examples of text before and after the annotation process is shown below:

### Example 1

**Original:** **Joey Muha** is a Canadian **drummer** from **Port Dover**, Ontario.
**Generalized:** [PERSON 1] is a Canadian [musician] from [town], Ontario.

### Example 2

**Original:** **Joakim Lindner** (born **22 March 1991**) is a Swedish footballer who plays for **Varbergs BoIS** as a midfielder. He is son to the competitive sailor **Magnus Olsson**.
**Generalized:** [PERSON 1] (born [date in the 1990s]) is a Swedish footballer who plays for [association football club] as a midfielder. He is son to the competitive sailor [PERSON 2].

## 4.2 Analysis

Table 2 details the level of generalization selected by the human annotators according to the entity type. Overall, only 36% of the selections landed on the default '***', meaning that a majority of text spans could be mapped to more meaningful replacement options.

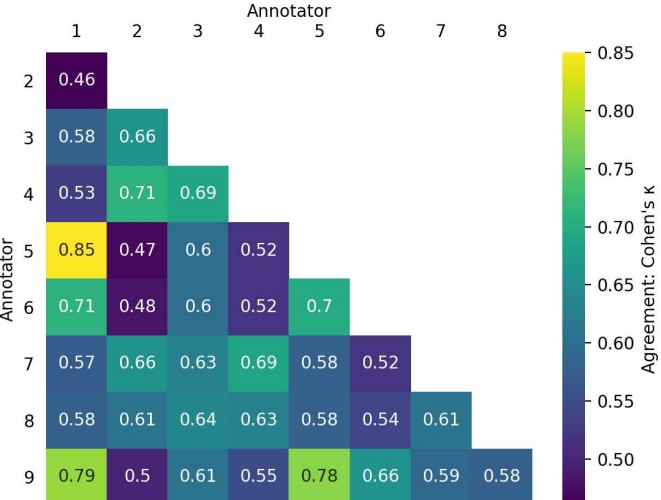

Figure 2: Pairwise agreement between annotators.

The generalization options were sorted from most specific to the most general following the hierarchical structure in Wikidata. Table 2 shows a clear preference for first level generalizations, meaning the annotators selected the least general option more frequently. It should be noted that some semantic categories (PERSON, QUANTITY) had fewer options. For instance, after manual inspection, we observe that 98 % and 96% of all selections made for QUANTITY and PERSON respectively are the least general option. For the MISC category, the corresponding percentage is only 20%.

A subset of the documents were annotated by multiple annotators. We estimated the inter annotator agreement using Light's kappa (L-kappa), as it allows annotators to select from a set of alternatives. It is computed as the mean of the Cohen's kappa of each annotation pair (Conger, 1980). A score of −1 indicates a direct disagreement, while 1 suggests perfect agreement. The L-kappa score obtained this data collection is 0.61, indicating a moderate to substantial agreement. Variations in the agreement between annotator pairs range from 0.46 to 0.85, as shown in Figure 2.

## 5 Conclusion

We presented an approach to generate replacements for detected PII based on heuristic rules and an ontology derived from Wikidata properties. The approach is employed to enrich an existing text sanitization dataset with suitable replacements for each text span. Those replacements were manually selected by annotators among a set of alternatives generated by the above approach.

The collected data highlights the benefits of this replacement strategy, with 64% of the text spans being mapped to a generalization other than the default '***'. However, the moderate inter-annotator-agreement also illustrates the difficulty of the task, which may admit multiple solutions.

Future work will focus on enriching the ontology, resolving entity linking ambiguities and using the dataset to train a neural model to select appropriate generalizations for PII spans.

## Acknowledgments

We acknowledge support from the Norwegian Research Council through the CLEANUP project (grant nr. 308904).

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

**Appendix A. Annotation guidelines**

The annotation guidelines describing the task, along with examples, are presented below.

# Replacement Choices in Text Sanitization: Annotation Guidelines

This annotation effort is part of a larger research project that seeks to understand how to automatically remove personally identifiable information from text documents (a problem called *text sanitization*). Personally identifiable information refers to any piece of information that may directly or indirectly reveal the identity of a particular individual. Text sanitization is an important problem when dealing with sensitive documents where we need to conceal the identity of given person(s) to protect their privacy.

The result of your annotation work will be included in a new, public dataset released under an open-source license.

## The Task

In this task, you are given a number of short biographies extracted from Wikipedia. To conceal the identity of the individual described in the biography, some text spans have already been marked as needing to be replaced. Each text span is shown in a drop-down menu where the values correspond to possible replacements. The original text span for which you will choose a replacement is also provided to help in the decision making process.

Your job is to select in each drop-down menu the best replacement for the text span according to the following two criteria:

1. The replacement should not disclose (directly or indirectly) the person's identity.

2. Provided that the above criteria is satisfied, the replacement should be as informative and readable as possible.

For example, in the sentence:

*PERSON 1 lives and works in **Oslo** ...*

possible choices for ***'Oslo'*** might include *[capital of Norway]*, *[city in Norway]*, *[city]* and *'***'*. The first choice is not general enough since it is as informative as the word Oslo. The second choice is more general, followed by the third choice, and finally the default *'***'*, which is least informative (but also least risky from a privacy perspective). Person names are by default replaced by *PERSON X* (where X is an integer).

## Procedure

The annotation work consists of the following steps:

- **Step 1** Read through the text once.

- **Step 2** For each marked span, look at the list of possible replacements and pick the most appropriate one. Only one replacement can be selected for each text span.

- **Step 3** When you are done with all replacements, review the text one final time. The selected replacements should not disclose the person identity, and the text should be as informative and readable and possible.

Many suggested replacements will be incorrect or irrelevant – this is normal and expected. If none of the suggested replacements are suitable for a given text span, you should choose the default '***' option.

## The '***' option

In all the dropout lists of possible replacements, there will be an '***' option. Use this if you find that no other replacement is appropriate.

Sometimes the '***' is the only suitable option, since you might encounter cases where the automatic generation of suggested replacements failed to come up with good options.

## Corner cases

There might be cases where a replacement looks appropriate but does not entirely fit the form of the sentence. For example, in the following sentence:

*PERSON 1 was born on* **May 18, 1943** *[...]*

The possible replacements will be *[1943]*, *[date in the 1940s]* and '***'. The most suitable choices in this case are *[1943]* and *[date in the 1940s]* (although it might necessitate some rephrasing to fit the current form of the sentence), not '***'.

## Example

Below you will find a step-by-step example of the annotation steps.

Start by briefly reading the text **(Step 1)**

Then for each of the spans choose one replacement **(Step 2)**. Following is a possible set of replacements chosen.

For example, the two decades could be replaced with the '***' option since they provide additional information along with the rest of the personal information still left in the text (e.g. *British, gay rights activist, general secretary* etc.) that could lead to the person being identified easier, which we wish to prevent.

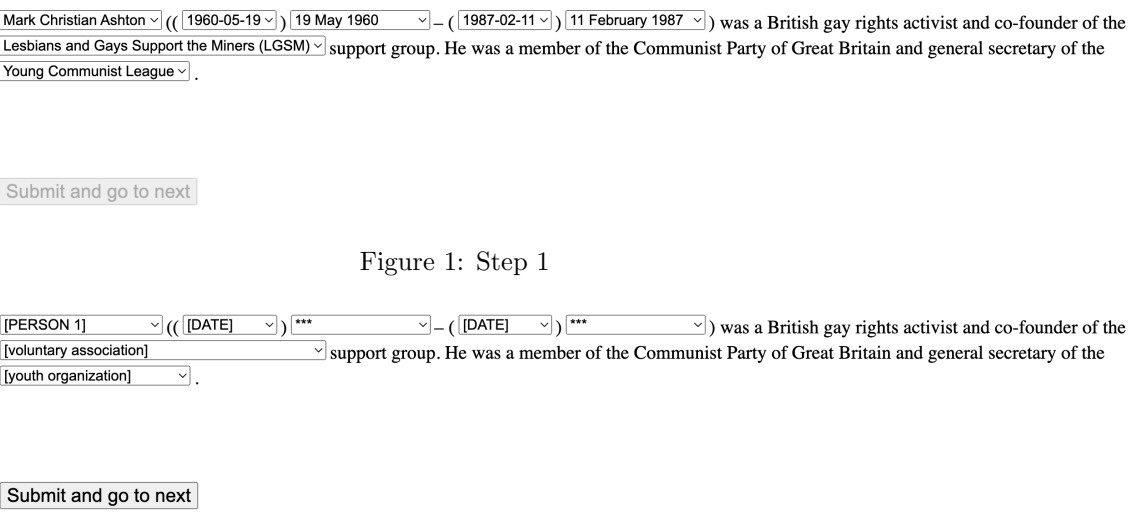

Figure 1: Step 1

Figure 2: Step 2

Note that there is no one correct solution, as long as the identity of the individual is not disclosed and the replacement choices result in an (as much as possible) informative text.

**NB!** You have to choose a replacement option. The original string is provided (first option in the drop-down list that cannot be chosen) in order to help choose the most appropriate one. The *Submit and go to next* button can only be clicked if replacements for all the spans have been selected.

Read the text with the selected replacements one last time **(Step 3)**. Make sure that you have chosen replacements for all text spans. Click on *Submit and go to next* to continue with the rest of the texts for this task.

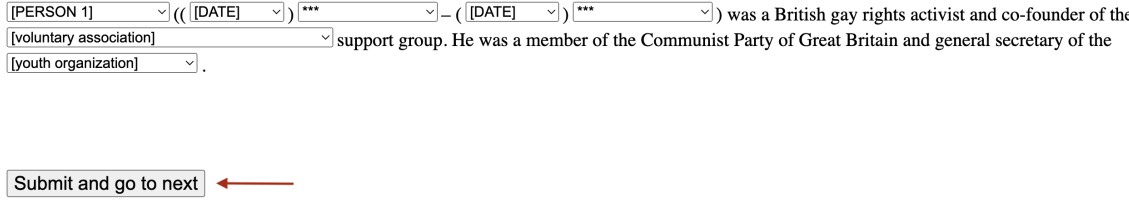

Figure 3: Step 3

A short message will appear on your screen when your assigned number of texts have been annotated.

