# OpenReview forum: "Generation of Replacement Options in Text Sanitization"
_NoDaLiDa/2023/Conference — NoDaLiDa 2023_

### Official Review · Reviewer_ZWjg · 2023-03-04
**Sound work, accept with minor revisions**

**Rating:** 7
**Confidence:** 4

**Review:**

The paper describes an approach to replace identified personal information with pseudonyms. The approach is sound, and any work that explores such approaches and spreads awareness of those is welcome.

The paper should cite work by Volodina et al where similar approaches are explored for Swedish:
* Elena Volodina, Yousuf Ali Mohammed, Sandra Derbring, Arild Matsson and Beata Megyesi (2020).  Towards privacy by design in learner corpora research: A case of on-the-fly pseudonymization of Swedish learner essays. In Proceedings of the 28th International Conference on Computational Linguistics (COLING) (pp. 357-369). (https://www.aclweb.org/anthology/2020.coling-main.32/)

For finding an appropriate pseudonym, you could consult an openly available multilingual resource GeoNames: https://www.geonames.org/
Please, add the column for total in Table 2.

Pseudonymizing by using "X" and "***" may introduce problems with automatic annotation of pseudonymized data. Have you thought of such consequences?

It would be interesting to hear a few words about how you plan to tackle other types of sensitive (and potentially disclosing) information, such as
* religious and political views, sexual orientation, medical conditions (e.g. in second Figure  in Appendix references to "gay",  "Communist party"), as well as
* false negatives like "Great Britain" and "general secretary" in the same Figure?


**Paper Type:**

Short paper

---

### Official Review · Reviewer_JuV4 · 2023-03-10
**text sanitization**

**Rating:** 5
**Confidence:** 4

**Review:**

The paper presents a method for text sanitization where personally identifiable information (PII) is replaced by more general information.
The result of this is an annotated dataset that contains sanitized Wikipedia articles.

There is one big problem with this research - it is not clear why the authors are doing this research. The authors do not say anything about the potential application of this sanitization tool. What kinds of text do they want to anonymize and for what purpose? The authors propose a generalization approach for PII replacements, but it is not clear whether this is a good idea at all. Although there might be applications, where such anonymization could be used, there are many situations, where such an approach would not be applicable at all.

Sanitization by itself is a bit subjective, but if there is no clear goal for it, then it gets very subjective. It is not clear how annotators could select the right generalization level if it is not clear for what purpose this anonymization is done.

The problem is highlighted by the text selected for annotation - Wikipedia. It is a public encyclopedia, there is no need to sanitize it, and any sanitization of it is useless and is difficult to imagine any application of such sanitized encyclopedia. If you want to build a meaningful sanitization solution, you should annotate texts that really need sanitization, and it should be very clear how sanitized texts are meant to be used. For example, court judgments, conversation histories, and medical data could be good candidates.



**Paper Type:**

Short paper

---

### Official Review · Reviewer_V5n8 · 2023-03-14
**Interesting short student paper extending on existing text anonymization work and data set.**

**Rating:** 7
**Confidence:** 3

**Review:**

**Quality** is good, with just a few minor typos
- Line 051 left: That do _no_ identify a person
- Line 132 left: NLP should be written out the first time, even though _we_ all know it's not Neuro-linguistic programming.
- Line 295 right: Drop "=" (or add it in the next parens as well, for symmetry)

**Clarity** is good, with concise explanation of what is done, and good examples.

**Originality** is questionable, as Carrel,  ..., Hirschman et al. seems to have done very similar things with a HIPS motivation 10 years ago.

**Significance** is hard to judge because of anonymous review, but will be good if the promised dataset is good and readily available.

Pros
- Well structured compact paper with Related Work, Generation, Dataset and Conclusion
- Good figures and tables
- Good evaluation with several annotators and properly calculated IAA Light Kappa-scores for a part of the texts.
- Interesting to see how annotators prefer as specific information as possible.
- Providing an enhanced dataset that others can presumably use.

Cons
- Hard to judge the quality of the new extended resulting dataset.
- Some of the listed examples seems a bit strange, like [association football position].
- Unclear exactly how the rule-based and the ontology-based division came about.
- Didn't understand the meaning of "incorrect" in
-- Line 278 "provided generalization options are incorrect". Who determines the incorrectness?

I would be interested in discussing the approach/data set further around a poster or in a Q&A-session, if I could join NODALIDA this year.
The Wikidata JSON.gz-file takes more than 10 hours to download, preventing further investigation for me today.
Sorry about submitting the review so late!

**Paper Type:**

Short paper

---

### Decision · Program_Chairs · 2023-03-17

Accept